# Root-Associated Microbiomes, Growth and Health of Ornamental Geophytes Treated with Commercial Plant Growth-Promoting Products

**DOI:** 10.3390/microorganisms9081785

**Published:** 2021-08-23

**Authors:** Gavriel Friesem, Noam Reznik, Michal Sharon Cohen, Nir Carmi, Zohar Kerem, Iris Yedidia

**Affiliations:** 1Department of Agroecology and Plant Health, The Robert H. Smith Faculty of Agriculture, Food and Environment, The Hebrew University of Jerusalem, Rehovot 7610001, Israel; gavrielf@volcani.agri.gov.il; 2Institute of Plant Sciences, Agricultural Research Organization, Volcani Center, Rishon Lezion 7528809, Israel; noamr@volcani.agri.gov.il (N.R.); michal.sharon78@gmail.com (M.S.C.); vhncarmi@volcani.agri.gov.il (N.C.); 3Department of Biochemistry, Food Science and Nutrition, The Robert H. Smith Faculty of Agriculture, Food and Environment, The Hebrew University of Jerusalem, Rehovot 7610001, Israel; zohar.kerem@mail.huji.ac.il

**Keywords:** microbiome, *Ornithogalum dubium*, *Pectobacterium*, perlite, soil mix, *Zantedeschia aethiopica*

## Abstract

The microbial community inhabiting a plant’s root zone plays a crucial role in plant health and protection. To assess the ability of commercial plant growth-promoting products to enhance the positive effects of this environment, two products containing beneficial soil bacteria and a product containing plant extracts were tested on *Zantedeschia aethiopica* and *Ornithogalum dubium*. The products were tested in two different growing media: a soil and a soilless medium. The effects of these products on *Pectobacterium brasiliense*, the causal agent of soft rot disease, were also evaluated in vitro, and on naturally occurring infections in the greenhouse. The growing medium was found to have the strongest effect on the microbial diversity of the root-associated microbiome, with the next-strongest effect due to plant type. These results demonstrate that either a single bacterial strain or a product will scarcely reach the level that is required to influence soil microbial communities. In addition, the microbes cultured from these products, could not directly inhibit *Pectobacterium* growth in vitro. We suggest density-based and functional analyses in the future, to study the specific interactions between plants, soil type, soil microbiota and relevant pathogens. This should increase the effectiveness of bio-supplements and soil disinfestation with natural products, leading to more sustainable, environmentally friendly solutions for the control of bacterial plant diseases.

## 1. Introduction

Ornamental geophytes include some of the most desirable cut-flower varieties in the industry [1,2]. Geophyte producers suffer significant losses due to soil-borne diseases, as well as bacterial soft rot disease caused by pathogens from the genus *Pectobacterium*, which macerate the plant tissue after they penetrate it and can spread rapidly through the whole plant [3,4]. Strict sanitation measures, as well as soil treatments with chemical compounds, have allowed the international trade in ornamental geophytes to flourish. However, many of these products have now been banned from commercial use, as they are considered harmful to the environment by the European Union, which accounts for the lion’s share of the trade in the ornamental industry [5,6,7]. Due to these restrictions, soil-borne diseases are an economic menace to geophyte growers, limiting their ability to produce top-shelf-quality cut flowers and bulbs for export. Growers are seeking “greener” solutions that will protect plants from pathogens, increase yields and protect the perennial bulbs year to year without the use of pesticides or harsh soil sanitizers. To date, there have been no in-depth studies of biological treatments for soil diseases in ornamentals [8]. Our goal was to examine the relationship between the underground portions of ornamental geophyte plants and the large microbial community that surrounds those organs. 

Plant roots experience the environment through the medium in which they reside. This could be a section of soil in a natural system, a soilless culture such as a hydroponic growth solution or inert minerals such as perlite [9,10,11]. The microbial community surrounding the roots plays a crucial role in the interaction between plant roots and the environment. The bacteria in the root-associated environment can protect the plant, increase its mineral uptake and exchange metabolites with the plant [12,13,14]. In recent years, due to regulatory restrictions and the rise of the biotech industry, there has been an increase in the availability of biocontrol and bio-stimulating products that contain soil bacteria and plant extracts. Some of these products are designed to trigger a systemic response in the plant and/or prime the plant’s defense mechanisms [15].

Certain microbes in the soil communicate chemically with the roots of different plants in a manner that is harmless or even beneficial for the plant. *Bacillus subtilis* is one such bacterium. Many strains of *B. subtilis* produce antibiotics, grow quickly and can populate many niches, increase plant iron uptake and contribute to soil health [16,17,18]. In light of this, many farmers and actors in the biocontrol industry are interested in using this sort of bacteria as an external supplement. However, the bacteria’s ability to communicate with the plant and elicit a positive effect such as defense priming or growth promotion can vary with the bacterial strain, plant species and environmental conditions. Plant-based products, which contain compounds that are known elicitors of different pathways in plants, such as silicates, have often been associated with such positive interactions. Silicates can prime the systemic acquired resistance (SAR) pathway, in which the plant tightens its defense system against pathogens and stress, mechanically, molecularly and physiologically [19,20].

In this study, we sought to describe the influence of different factors on root–soil–microorganism interactions in terms of the population dynamics of the microorganisms in the root-associated environment. In this work, we used two different growing media, a planting mix rich in organic matter and the inert mineral perlite, each planted with two geophyte species, *Zantedeschia aethopica* and *Ornithogalum dubium*, and regularly treated with commercial biocontrol products. We hypothesized that the use of the biocontrol products would be associated with altered composition of soil bacterial communities. The results were in line with previous reports, relating the microbial populations of the root-associated environment to the growing media and the plant species. 

## 2. Methods and Materials 

### 2.1. Biological Material and Growth-Promoting Products

Bulbs of two ornamental geophytes, *Z. aethiopica* and *O. dubium*, were purchased from local producers for this research. The same *Z. aethiopica* bulbs (cv. ZAI) were used throughout the entire study. Fresh *O. dubium* (cv. M13) bulbs were bought each year. Three commercial growth-promoting products were assessed for their ability to influence the population dynamics of the root-associated microbiome. One product, Agriotics© made by Ecosense (Rishon Lezion, Israel), was a liquid solution of *Bacillus subtilis* spp. The manufacturer claims that this product has a probiotic effect on soil and plant health. Another microbial product applied was Rhizoctol^®^ produced by Adama (Beer Sheva, Israel), a spore powder of *Bacillus amyloliquefaciens* (strain FZB42), which has been found to be effective against fungal soil pathogens in potatoes and is considered to be antagonistic to *Rhizoctonia solani* [21]. The third product was a plant extract-based detergent, GreenUp Soil^®^ by Green Life Group (Ashdod, Israel), used for natural soil disinfestation, in addition to growth promotion. This product contains unique metabolites and silicates. *Pectobacterium brasiliense* strain Pb3, the causal agent of soft rot, which was isolated from potato fields in Israel during 2008 [22], was used for infection assays. *Escherichia coli* (*E. coli*) K12 was used to calibrate the 16S rRNA primers and the genomic DNA extraction. All soft rot disease incidents in the greenhouse or on the bulbs occurred from natural infestation of the bulbs or the growing media. 

### 2.2. Bacterial Growth Media

The bacterial strains were cultivated at 28 °C in lysogeny broth (LB; Difco Laboratories, MI, USA) or plated for dual culture assays on minimal medium (MM) prepared as described previously [23].

### 2.3. Growing Conditions

During 2018, plants were grown in containers filled with perlite (Agrekal, Habonim, Israel). The containers were 5 m long, 80 cm wide and 30 cm deep. There were 24 containers for each plant species, bulbs were sown in two lines, 17 cm apart, 45 plants per tank, with the plants grown in a polyethylene-covered greenhouse during the winter season (5–25 °C), under natural light/temperature conditions. Each treatment included six random repeats, including a tap-water control. In 2019, plants were grown in smaller containers (104 cm long, 56 cm wide and 20 cm deep) under similar conditions, with the same number of containers and repeats per treatment as in the first year, with 10 plants per tank. To study the effect of the growing medium on the richness and stability of the bacterial population around the root, in 2019, the tanks were filled with a planting mix purchased from Tuf Marom Golan (Golan Heights, Israel) that was enriched with organic matter in the following ratios (by volume): peat moss, 20%; coconut fiber, 70%; and compost, 10%. The bulbs were soaked in the treatment solutions for a period of 15 min before planting and their leaves and other aboveground organs were treated manually by overhead watering every 2 weeks throughout the growing season, according to the manufacturers’ instructions (GreenUp, 0.05%; Ecosense, 0.01%; Rhizoctol, 0.01%). Plant growth and flowering parameters were monitored throughout the growing season to evaluate the effects of the treatments. In both years, bulbs were planted in the last week of September and were grown until the last week of June, when the bulbs were harvested and their growth and health parameters were evaluated. For health parameters, the severity of naturally occurring soft rot disease in tubers/bulbs per treatment was assessed using the following index: (1) completely healthy; (2) slightly soft; (3) watersoaked spot of soft rot; (4) half of the bulb is rotten; and (5) bulbs are completely rotten. For growth parameters, the amount of new bulbs and cumulative weight of bulbs were measured.

### 2.4. Molecular Analyses

Soil samples were collected at the end of the flower-picking season (mid-April) by filling 50 mL tubes from the center of each growing container, avoiding any shearing of roots and rhizomes. Samples were stored at −20 °C until DNA extraction. DNA was extracted using three different methods: a manual, phenol-based method and two commercial kits designed for genomic DNA extraction aimed at amplification of the prokaryotic 16S ribosomal RNA gene. The two commercial kits were DNeay Power Soil (QIAGEN, GmbH, Hilden, Germany) and Nucleospin Soil (Marchery Nagel, Duren, Germany). The phenolic process was conducted as described previously [24]. Briefly, PBS, TNC and phenol were mixed with samples and glass beads and subjected to three rounds of bead-beating and centrifugation. Phenol-chloroform-isoamyl alcohol was added to the lysates, the mixtures were centrifuged and each supernatant was mixed with glycogen and a 30% solution of PEG 8000. After 1 h of centrifugation, the pellet was mixed with 75% ethanol and centrifuged twice, dried and eluted in molecular-grade water. DNA extraction using the kits was performed according to manufacturer instructions. A fragment of the 16S rRNA gene containing the V4 and V5 variable regions was amplified using gene-specific primers: 16S rRNA V4-5 construct 515F-926R [25]. Illumina target-specific linker sequences, cs1 or cs2, were included in addition to the 16S rRNA gene-specific sequences, which allowed sample indexing and pooling [26,27]. 

Each PCR amplification was conducted using a TIGER Taq mix PCR kit (Hylabs, Rehovot, Israel) in a final volume of 25 μL per reaction, according to the manufacturer’s instructions. PCR products were tested for fragmentation and band density on a 1.8% agarose gel, by electrophoresis for 40 min at 100 V. Once the best extraction procedure was determined, PCR products were sent to sequencing. Amplicon libraries were sequenced on an Illumina MiSeq instrument (Illumina, San Diego, CA, USA) at the University of Illinois Sequencing Core (Chicago, IL, USA). Sequencing was conducted in paired-endmode (2 × 300 bp) with the use of a v.3 (600 cycles) chemistry cartridge, which allowed the generation of long paired reads fully covering 16S V4–V5 amplicons. 

Microbiome analysis was performed at the Plant Pathology Department, Volcani ARO (Beit Dagan, Israel) based on Fast.q files, using the Qiime2 pipeline [28] with a cutoff of 97% identity. A sequencing depth of 11,000 sequences was obtained using DADA2 [29] and Demux plugins [28]. The data were processed and analyzed for weighted and unweighted UNIFRAC, Shannon index and Faith-PD group of significance, with a permutational multivariate analysis of variance (PERMANOVA; *p*_v_ = 0.0001) for assessing the α and β diversity of the treatments.

### 2.5. Dual-Culture Assays

Growth-promoting products containing *B. subtilis* and *B. amyloliquefaciens* were tested for their interaction with the bacterial plant pathogen Pb3 in vitro. A single colony of Pb3 was grown overnight at 28 °C in liquid culture. In the morning, bacteria (CFU = 1 × 10^7^) were streaked in a straight line, about 2 cm from the center of the plate. The same was done with both plant growth-promoting *Bacillus* products, which were incubated at their respective concentrations for 6 h at 30 °C. Each *Bacillus* was plated in parallel line to Pb3 on the same plate or double-distilled water as a negative control. Once dried, the plates were incubated for 72 h, scanned and observed for qualitative evaluation of the interaction. Each experiment was conducted in three replicates and in two separate experiments, with similar results. The results presented are one representative plate of one experiment.

### 2.6. Statistical Analysis

All data were analyzed with JMP-Pro software, version 13.0 (SAS Institute Inc., Cary, NC, USA) using one-way analysis of variance (ANOVA). When ANOVA indicated significance (*p* ≤ 0.05), Student’s *t*-test was performed to test each treatment relative to the water control. Graphs were generated using Excel16 (Microsoft, Redmond, WA, USA).

## 3. Results

### 3.1. Root-Associated Microbiome Analysis 

Soil samples were collected each year from the greenhouse for microbiome analysis and to compare the root-associated microbiomes of the different plants and growing media. In the first season, the plants were grown in perlite, an inert mineral medium whose ability to host diverse or selective bacterial communities and compatibility with efficient nucleic acid extraction have not been previously examined. Accordingly, DNA-extraction protocols were calibrated and the quality of sequencing depth was compared to that of the data from the planting mix, a medium rich in organic matter that was used in the second season.

The comparison of extraction methods revealed that the high-yield phenolic method did not yield a sufficient amount of gDNA from perlite and could not purify the planting mix samples adequately, as shown in Figure 1. The standard kit DNeasy Power Soil was not as fine-tuned and productive as the Nucleospin Soil kit by MN; this was especially true for the perlite samples. The Nucleospin samples gave the best extraction results and the materials extracted from those samples were sent for sequencing.

According to greenhouse observations based on the development of naturally occurring soft rot disease symptoms, the presence of soft rot was higher in the first year (perlite) and lower in the second year (planting mix). We wanted to try and identify variability in the presence of this pathogen in the soil through taxonomic annotation (Appendix A). The results of that examination revealed the presence of bacteria from the *Enterobacteriaceae* family (to which *Pectobacterium* belongs) in all of the perlite treatments, with the highest relative rate found for the water × perlite treatment, accounting for 0.6% of the total operational taxonomic units (OTUs) sequenced in these samples (Figure 2). In the second year, low rates to none (0–0.12%) of *Enterobacteriaceae* bacteria were found in the planting mix.

During the second year, we examined the differences between the population dynamics of the microorganisms seen previously in the inert growing medium (perlite) to those of the microorganisms in a medium richer in organic matter (planting mix). It is important to note that the planting-mix samples yielded more sequence groups (OTUs) than the samples taken from the perlite medium, as presented in Figure 3.

The water × *O. dubium* × perlite samples had the least OTUs of all of the different treatments. The combination of the planting mix with the water control treatment had the largest number of OTUs out of all the samples in both growing seasons. 

A principal-of-coordinates analysis (PCoA) that was carried out to measure proximity between populations (Figure 4) revealed that Axis 1 constitutes 17.9% of the explained variance between growing media, while Axis 2 constitutes 7.3% and Axis 3, 4 and 5 constitute about 4.5% each. All of the results from the planting mix are clustered above Axis 1, indicating greater variability between the samples from this medium. The data from the perlite medium were clustered between Axes 2 and 3, indicating smaller differences between those samples.

As can be seen from Figure 4A, the strongest effect on the bacterial population in the root-associated environment was due to the growing medium. An additional difference was observed between the two plant species during the second growing season (Appendix A). Specifically, there was a clustering of plant species in the planting-mix cluster (Figure 4B), while no differences were observed during both growing season between any of the growth-promoting treatments. The analysis of each year separately in unweighted Unifrac revealed that plants “determine” the microbial population in the root-associated environment (Figure 5), depending on the medium, apparently in association with the ability of the medium to support microbial diversity. In the planting mix, a significant difference was noted between the microbiomes surrounding *O. dubium* roots and those surrounding *Z. aethiopica* roots. In the perlite, clustering by plant species was noticeable, but not significant. Additionally, in all of the analyses, we found no significant differences between the growth-promoting products when we controlled for the plant species and the growing medium. Interestingly, in the weighted Unifrac analyses, no significance difference was found between the two plant species in the planting-mix medium (Appendix A). Still, data from both years reveal an effect of plants in shaping microbial communities in the root-associated environment. It is notable that in the taxonomic screening of the microorganisms found in the perlite medium, we found a higher relative abundance of *Bacillus* genus in the microbial treatments than in the control or plant extract treatment. No such effect was observed in the planting mix.

### 3.2. Dual-Culture Assay

This experiment was performed under in vitro conditions, in order to test for possible antagonistic effects between *Bacillus subtilis* or *Bacillus amyloliquefaciens,* the microbial constituents of the growth-promoting products (Ecosense and Rhizoctol) and Pb3 on a solid medium. The average area covered by the *Bacillus* spp. or Pb3 was evaluated at 24 and 72 h after the application of the bacteria. A representative plate from each treatment or control at 72 h post-application is presented in Figure 6. Although no significant inhibition was observed for any of the products, the results could point out a possible inhibition of the *Bacillus* strains by Pb3, as well as accelerated growth of Pb3 in the presence of *Bacillus*, as compared to its rate of growth on the control plates (Figure 6). Following 72 h, accelerated growth of Pb3 in the presence of the *Bacillus* strains was significant with mean colony areas of 5.29 cm^2^ and 5.47 cm^2^ when grown across *Bacillus subtilis* from Ecosense (*p*_v_ = 0.06) and *Bacillus amyloliquefaciens* from Rhizoctol (*p* = 0.04), respectively, as compared to a mean area of 4.3 cm^2^ for colonies of Pb3 grown in the water control. After 72 h on the control plates, the area covered by *Bacillus subtilis* from Ecosense (7.98 cm^2^) was significantly larger than the area covered by the same bacteria in the presence of Pb3 (5.76 cm^2^), and significantly larger than the area covered by *Bacillus amyloliquefaciens* from Rhizoctol on the control plates (5.39 cm^2^). According to the results, there is no antagonistic inhibition of Pb3 by the *Bacillus* spp. found in either of the tested products.

### 3.3. Geophytes’ Responses to the Growth-Promoting Products

Under the experimental conditions, none of the examined treatments significantly promoted plant growth or reduced naturally occuring soft-rot symptoms in *Z. aethiopica* or *O. dubium*, in comparison to a water control. During the first year, in *Z. aethiopica*, none of the treatments induced any significant changes in germination rate, leaf number, plant height, number of flowers or bulb weight (Figure 7). During the second year, a significantly greater number of flowers were seen in the water control treatment (*n* = 63) than in the Ecosense (*n* = 43) or Rhizoctol (*n* = 32) treatments. Evaluation of the bulbs after the second year revealed significantly heavier bulbs in the GreenUp treatment, as compared with the control or other treatments. 

During the first growing season, in *O. dubium*, growth inhibition and reduced flowering were observed in response to the bacterial treatments and a high rate of naturally occurring soft rot disease was observed in all treatments (Figure 8C). Although the highest disease level was measured in the water control treatment, this level of disease was not significantly different from the levels observed for the other treatments. During the second year, *O. dubium* exhibited higher disease levels in the experimental treatments rather than the control, with the GreenUp treatment associated with a significantly greater incidence of disease (as presented in Figure 8D). However, the GreenUp treatment had a positive effect on propagation, with a significantly larger amount of new bulbs than the control (Figure 8B).

## 4. Discussion

The influence of the environment on intimate interactions between microbes and plant roots has been previously established [30,31,32]. Our results suggest that the root-associated microbiomes of two ornamental geophytes are determined by the unique interactions between each geophyte species and its microbiome, as demonstrated in two different environments. As a mineral substrate, perlite-based medium is known for its restriction of biological richness, as compared to media that are rich in organic material [11]. The microbiome of plants grown in that medium supports this, as evidenced by the lower number of OTUs and higher relative abundance of *Enterobacteriaceae* in that medium. The microbiome analysis of the root environments of *Z. aethiopica* and *O. dubium* provided a fascinating look at factors that influence the diversity and richness of the bacterial community in the root-associated environment. As far as we know, this is the first report on the variability of the composition of microbial populations in the root environments of different flowering geophytes grown under similar conditions. 

In 2018, perlite served as the growing medium. At the beginning of the study, no reports were available on plant microbiome studies in which perlite had been used as a growth support. Perlite is an inorganic mineral that is considered inert without any presence of organic matter. Our hypothesis was that under such inert conditions, the effects of beneficial treatments that support microbial growth would be more substantial. As the results of the first year did not yield any significant advantage for the growth-promoting treatments, in the following year (2019) we decided to focus on a growing substrate that contained higher levels of organic material and aimed to increase the richness and diversity of microbial populations. Based on previous microbiome studies, we chose to use a common planting mix that contained 10% compost as a source of organic matter. This allowed us to compare the microbiomes of the two geophytes in different growing media [33,34,35,36,37]. Previous plant microbiome studies, which examined differences in the population of microorganisms, revealed that geographical location, soil type/substrate and plant species are the factors that have the strongest influence on microbial population (in that order) [10,38,39].

At the plants’ early-bloom stage in the soil microbiome studies [40], sequencing procedures did not meet the standard of at least 10^4^ sequences per sample [41]. These limited observations revealed differences between plant species. However, those differences were not significant. Working with perlite as a growing media presented another challenge: there was limited information in the literature regarding DNA extraction from this material, which has not been necessarily developed for bacterial DNA [24,42,43]. It seems that comparisons between different methods for DNA extraction from inert substrates are uncommon in the literature, especially in the context of microbiomes. 

Our results revealed that the perlite samples had significantly fewer OTUs than samples collected from the planting mix, regardless of the treatment or plant species. This result indicates that the root environment in the perlite substrate is not supportive of bacterial growth [44], and as a result the quality of extracted DNA for the subsequent sequencing reaction was lower, compared to the substrate containing organic matter. The results support the choice of a growing medium to increase species richness and the amount of microbiotic interactions in the substrate. It is likely that the initial ‘bacterial load’ in the perlite medium was substantially lower than that of the planting mix substrate, and was thus responsible for the difference in “competitive exclusion” capacities of the two media. In literature there has been evidence that increasing the amount of organic material in the substrate may positively affect plant health and reduce pathogenic interactions [33,45,46]. Microbiome assessment consists of a large number of statistical analyses, designed to distill quantitative differences from the huge amount of genetic sequences obtained in the processing of the samples. The challenge in these analyses is to give scientific significance to these differences before application [28,47].

Here, we aimed to elucidate every relationship between bacterial populations and the plant species, the growing media, the beneficial treatment and soft-rot disease development during each of the growing seasons. Connections between bacterial populations (revealed by the microbiome analysis) and greenhouse observations could be revealed. For example, a lower relative rate of bacteria from the *Enterobacteriaceae* family, to which *Pectobacterium* (the causal agent of soft-rot disease) belongs, was found in the planting mix, as compared to the perlite substrate. Apparently, the elevated competitive environment in the planting mix reduced the relative abundance of the pathogen in this medium. Since bacterial load was not determined, we could only refer to the relative abundance of the different OTUs. The occurrence of soft rot symptoms was higher in perlite, as observed in the greenhouse and on the bulbs, suggesting that a microbiome that contains little organic matter may be less suppressive to pathogen development [45]. Moreover, in perlite culture, lower rates of *Enterobacteriaceae* OTUs were found in *Z. aethiopica* plants than *O. dubium* plants (Figure 2). *Z. aethiopica* is known to be more tolerant to soft rot disease than *O. dubium* [48,49,50]. 

The PCoA analysis examining the proximity of bacterial populations in different samples was performed without species abundance, expressed as the number of sequences associated with the same taxonomic unit (OTU), so that the results could reveal differences in unique species present at low levels (i.e., plant symbionts, substrate residents, treatment bacteria and disease agents) [10]. This type of PCoA is called an unweighted Unifrac. The results of this analysis provide better insight into the separation of the different treatments between soil type and plant species. When species abundance was taken into account, in the weighted Unifrac analysis, greater microbiome diversity was noted for the planting-mix medium and the results for the different repetitions of the water control treatment segregated by plant species. The differences observed between treatments may indicate that some of the bacterial treatments were able to moderately affect the bacterial presence near the roots, as can be seen in the abundance of each OTU found. However, despite some clustering of the different treatments, there were no significant differences between the treatments in any of the plants or substrates. If any of the treatments had been thriving in one of the substrates or plants, the results of both analyses would support this [30]. These results emphasize the need to explore the efficacy of any beneficial bacterial product for a particular combination of plant species and growing medium. 

The large differences between the growing media shown here are widely recognized [38,51,52]. Bacterial sequences associated with the taxonomic family of *Pectobacterium* were found. The levels at which this family was found in 2018 are of the same order of magnitude as the levels of members of the taxonomic family of the two *Bacillus* treatments (<1%) supplemented to plants regularly, and can establish a baseline for the presence of *Pectobacterium* that can cause disease. 

To summarize, the highest correlation was shown for the growing media and it may be concluded that the growing medium has the strongest effect on the composition of the microbiome. The second most influential factor was the species of the plant: *Z. aethiopica* or *O. dubium*. The growth-promoting treatments did not significantly affect the composition of the microbiome. The planting mix containing organic matter was found to be richer in bacteria and it was easier to sequence the DNA of those bacteria than that of the bacteria present in the perlite.

Two different *Bacillus subtilis* products were applied here as unique formulations of commercial products. i.e., Ecosense and Rhizoctol. Both of these products were tested on the fungal pathogens *Rhizoctonia* in potatoes and *Fusarium* in tomatoes, both in real soil grow-ops, and were deemed successful, as claimed by the manufacturer. The Ecosense treatment, which is considered a probiotic treatment [15], showed inhibited growth in the presence of the plant pathogenic bacteria Pb3, while Pb3 expressed accelerated growth on those same plates. Thus, the Ecosense treatment was not found to inhibit the growth of Pb3 bacteria even though bacteria from this treatment grew to cover the largest area at any given time under control conditions. Bacteria need a favorable soil environment in order to strengthen the host plant by indirect mechanisms such as increasing nutrient availability, eliciting plant resistance and competing with other soil microorganisms, as proposed in previous studies [53,54]. Here, the bacterial treatments were unable to directly prevent the growth of Pb3, a naturally occurring agent of soft rot disease in geophytes. Another possibility is that Pb3 produces an antibiotic that may inhibit *Bacillus* bacteria, in line with the ability to produce the carbapenem antibiotic that has been reported for some pectobacteria [55].

The geophytes *Z. aethiopica* and *O. dubium* were grown in different growing media in each growing season (2018, 2019) and treated with growth-promoting products (bacteria or a natural extract containing silica) throughout the growing season. It had been assumed that the lack of biological richness in the perlite-based medium may enhance the effects of the treatments [56]. Previously, consistent and continuous fertilization with chemical fertilizers was found to dramatically change the nutrient levels and soil pH, which gives an advantage to certain bacterial species in the soil [57,58]. Here instead, the disease-causing bacteria were established much better than the beneficial bacteria and no effects of the beneficial bacteria on the plants were observed (as observed by evaluation of soft rot symptoms), raising the question if they were present at all. 

## 5. Conclusions

This study examined the growth-promoting effects of commercial products that are considered environmentally friendly, and the abilities of those products to control soft rot in flowering geophytes. Testing three commercial products in two planting substrates that are commonly used for the cultivation of ornamental geophytes, it was shown that the composition of the microbiome was hardly affected by the presence of the beneficial microbes or the different treatments. It may be concluded that environmental conditions, especially in the presence of high levels of naturally occurring soft rot disease caused by *Pectobacterium* infestation, will adversely affect the efficacy of supporting bacterial treatments in an inert substrate. 

Further research will allow the development of methods to establish beneficial and engineered soil communities by the use of plant-growth promoters for protecting crops. The observed resemblance between the treatments, in terms of the examined parameters or desirable plant phenotypes, points to the growing medium being the strongest influence on the microbiome communities. Planting-mix medium is shown to be superior for both tested geophyte crops in terms of both the richness of the bacterial communities and the repression of soft-rot disease, as observed in the microbiome analyses and soft rot disease assessment in the greenhouse and in bulbs, respectively. It is concluded that the microbial diversity in the substrate is largely responsible for the stability of the microbiome, and the greater diversity contributes to greater stability in the face of variable environmental conditions. This work underscores the need for a better understanding of the multiple interactions between the bacteria found in the vicinity of plant roots, the soil and the plants. Assessment of microbial densities in the soil or growth media prior to planting and understanding of other components of the microbiome (including fungi, protozoa and archaea) would improve the understanding of the overall robustness of competition encountered by a pathogen. Improving our understanding of these interactions and of the modes of action of plant growth-promoting products can greatly promote their field applications. 

## Figures and Tables

**Figure 1 microorganisms-09-01785-f001:**
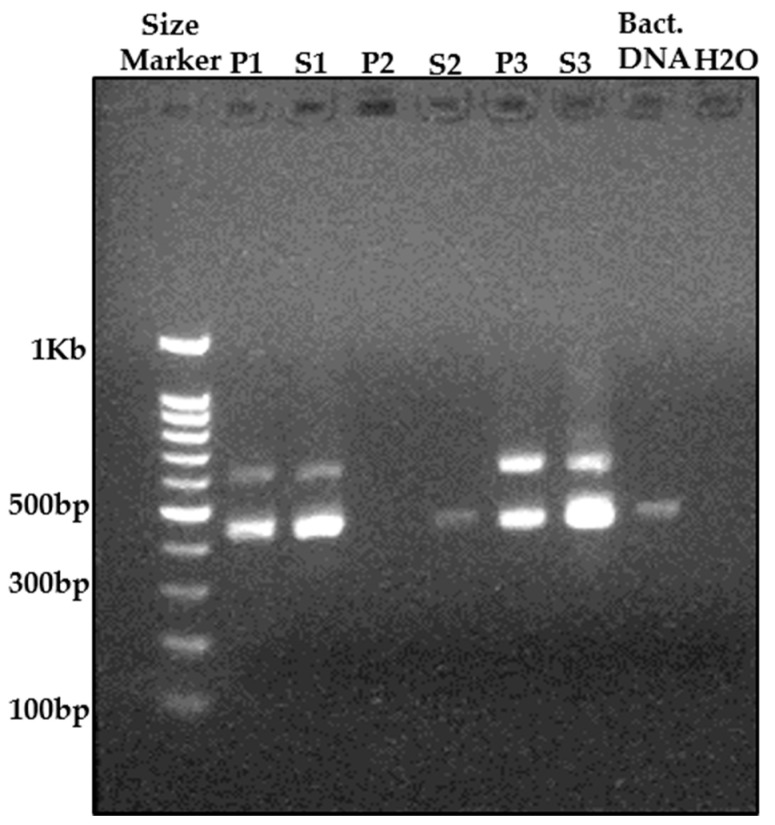
Agarose gel electrophoresis of 16S rRNA gene V4 & V5 regions of PCR products to compare different methods of extracting genomic DNA from soil bacteria. The different combinations of growing media and extraction methods are as follows: P—Sample collected from *Ornithogalum dubium* grown in perlite during the first growing season. S—Sample collected from *O. dubium* grown in planting mix in the second growing season. Bact. DNA—Positive control, mixture of *E. coli* k12 and *Pectobacterium brasiliense* (Pb3) mixed with planting-mix substrate. H_2_O—used as negative control for the PCR reaction. The extraction methods are numbered: 1—Phenolic extraction protocol. 2—Extraction with C-TAB only. 3—MN Nucleospin Soil Kit. The samples were taken from plot DA3 (repeat #3, water control treatment).

**Figure 2 microorganisms-09-01785-f002:**
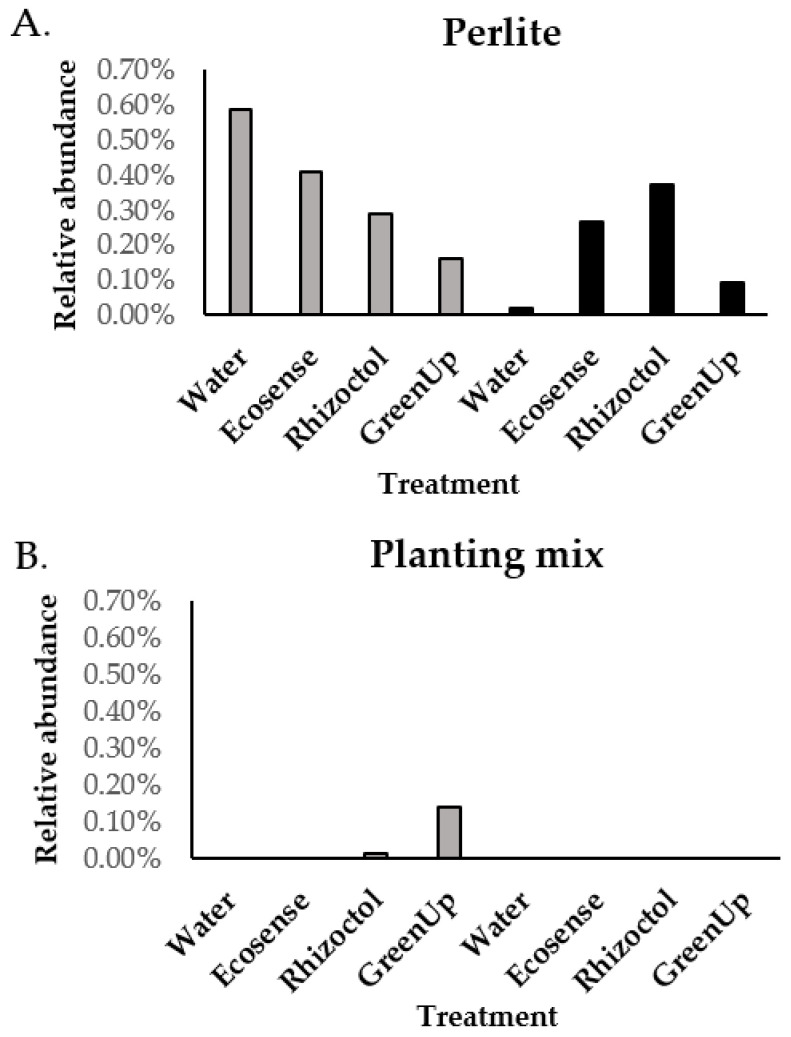
Presence of *Enterobacteriaceae* bacteria in the root environments of *Ornithogalum dubium* and *Zantedeschia aethiopica* in two growth media. Columns represent the relative frequency of DNA sequences identified as belonging to the family *Enterobactriaceae* in samples of soil surrounding the roots of *O. dubium* (gray) and *Z. aethiopica* (black) in (**A**) perlite and (**B**) a planting mix. There were three replicates of each treatment.

**Figure 3 microorganisms-09-01785-f003:**
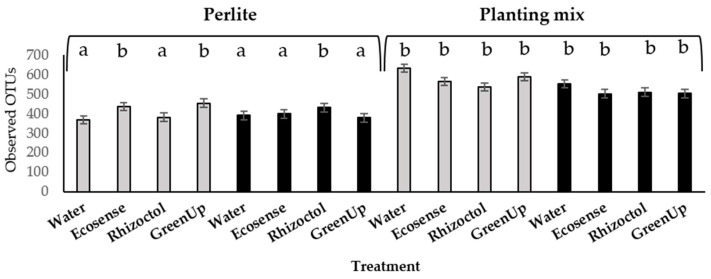
Mean operational taxonomic units (OTUs) per treatment, identified in the root-associated environments of *Ornithogalum dubium* and *Zantedeschia aethiopica* in two growth media. Columns represent mean numbers of OTUs in samples taken from three repetitions of each treatment of *O. dubium* (gray) and *Z. aethiopica* (black) grown in perlite or planting mix. Bars represent ± standard errors. Letters represent significantly different statistical groups.

**Figure 4 microorganisms-09-01785-f004:**
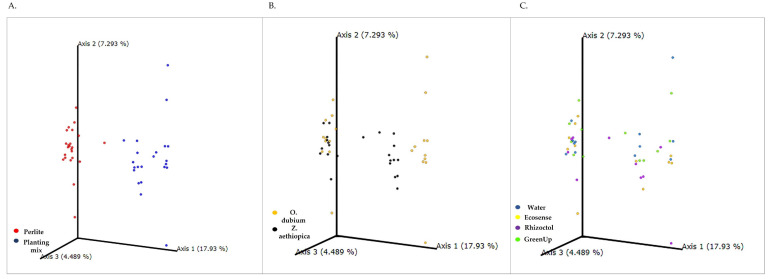
Principal-coordinates analysis (PCoA) plots of unweighted Unifrac distance metrics of bacterial populations in the root-associated environment of *Ornithogalum dubium* and *Zantedeschia aethiopica* based on data from two growing seasons. (**A**) PCoA plot of data for the two growing media: perlite (red) and planting mix (blue). (**B**) PCoA plot of data by plant species: *O. dubium* (orange) and *Z. aethiopica* (black). (**C**) PCoA plot of data by growth-promoter treatment: water (control; blue), Ecosense (yellow), Rhizoctol (purple) and GreenUp (green). Each data point represents sequences from a single soil sample; the plot contains all of the samples collected during two growing seasons.

**Figure 5 microorganisms-09-01785-f005:**
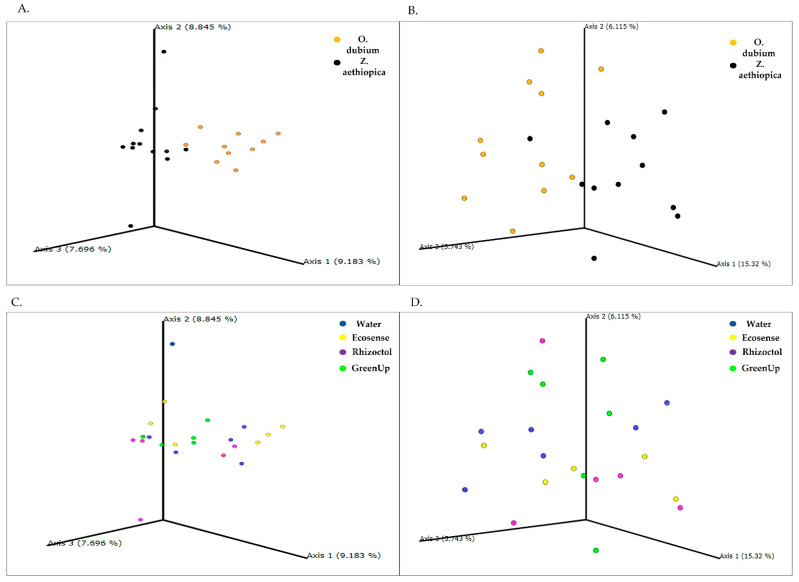
Principal-coordinates analysis (PCoA) plots of unweighted Unifrac distance metrics of bacterial populations in the root-associated environments of *Ornithogalum dubium* and *Zantedeschia aethiopica*. Data for each year were analyzed separately. (**A**) The different plant species in perlite: *O. dubium* in orange; *Z. aethiopica* in black. (**B**) The different plant species in the planting mix: *O. dubium* in orange; *Z. aethiopica* in black. (**C**) Different growth-promoter treatments applied to plants grown in perlite: water (control; blue), Ecosense (yellow), Rhizoctol (purple) and GreenUp (green). (**D**) Different growth-promoter treatments applied to plants grown in planting mix: water (control; blue), Ecosense (yellow), Rhizoctol (purple) and GreenUp, (green). Each data point represents sequences from a single soil sample; each plot contains all of the samples collected during that growing season.

**Figure 6 microorganisms-09-01785-f006:**
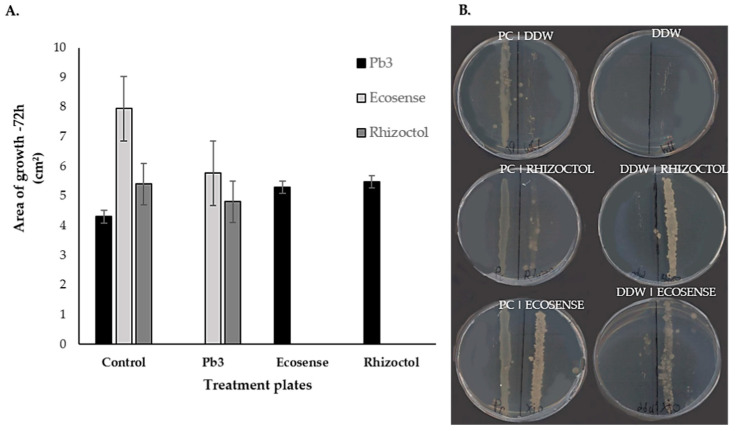
Differences in the area populated by *Bacillus* and *Pectobacterium brasiliense* (Pb3) on solid LB agar. *Bacillus subtilis* from Ecosense or Rhizoctol products were plated on LB plates by spreading a 10 μL droplet, or 10 μL of Pb3 at a concentration of 1 × 10^8^ CFU/mL or 10 μL of sterilized water. There were five replicates of each combination of *Bacillus,* Pb3 or water. (**A**) Comparison of the average area covered by each bacterial treatment (Pb3/Ecosense/Rhizoctol) or water (negative control) on a Petri dish, as measured at 72 h post-inoculation following incubation at 28 °C. The bars represent the standard errors for each treatment. Statistical variance was tested by a *t*-test at a significance level of *p* < 0.05. (**B**) Representative images of the experiment plates at 72 h post-inoculation. Left column (top to bottom): Pb3 vs. water (DDW); Pb3 vs. *B. amyloliquefaciens* (Rhizoctol); Pb3 vs. *Bacillus subtilis* (Ecosense). Right column (top to bottom): water (DDW) only; *B. amyloliquefaciens* (Rhizoctol) vs. water (DDW); *B. subtilis* (Ecosense) vs. water (DDW).

**Figure 7 microorganisms-09-01785-f007:**
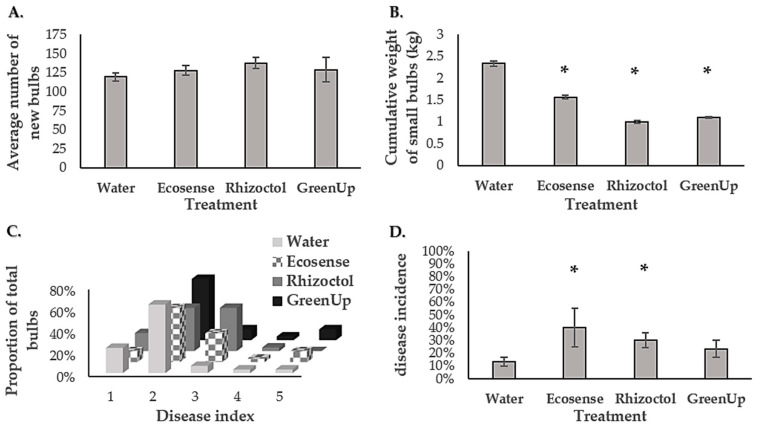
Propagation indices and damage from soft rot in *Z. aethiopica* tubers after the first growing season in perlite in the greenhouse. (**A**) Average number of new tubers formed during the season around the planted tubers in each treatment plot. (**B**) The cumulative weight of all small (less than 15 mm in diameter) tubers at the end of the growing season for all repeats in each treatment. (**C**) Severity of naturally occurring soft rot disease in tubers per treatment assessed using the following index: (1) completely healthy; (2) slightly soft; (3) stain of soft rot; (4) half of the bulb is rotten; and (5) completely rotten. (**D**) Average disease incidence for each treatment (3–5 in the disease index). Bars represent standard errors. An asterisk above the bar signifies a significant difference relative to the control treatment (Student’s *t*-test, *p* < 0.05).

**Figure 8 microorganisms-09-01785-f008:**
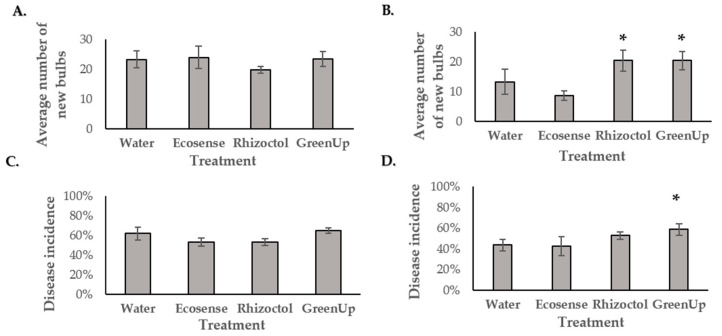
Propagation indices and damage from soft rot disease in *Ornithogalum dubium* bulbs after each growing season. Average number of new bulbs formed around the mother bulb during the season (**A**) in perlite and (**B**) planting mix. Incidence of soft rot disease (**C**) in perlite and (**D**) planting mix. Naturally occurring soft rot disease incidence was calculated as the average percentage of bulbs showing clear signs of rot (3–5 on the disease index) from all of the replicates for each treatment. Bars represent standard errors. An asterisk above the column signifies a significant difference relative to the control treatment (Student’s *t*-test, *p* < 0.05).

## Data Availability

The data presented in this study are available on request from the corresponding author.

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
