# Peer review of "Root-Associated Microbiomes, Growth and Health of Ornamental Geophytes Treated with Commercial Plant Growth-Promoting Products"

_microorganisms, 2021, doi:10.3390/microorganisms9081785_

Round 1

Reviewer 1 Report

Overall, this manuscript was a pleasure to read.  It is a well-organized and clearly written document.  The value of bio-stimulants and microbial inoculants as a solution for the control of plant diseases or enhanced plant nutrition is a recurring question.  As such, this work directly addresses that question and is one of the few studies willing to report the lack of efficacy…even though the vast majority of trials have demonstrated ‘failure’ but with the caveat that ‘additional study’ will lead to success.  Likewise, the report of a lack of effect of the inoculant on the soil microbiome is of value and again demonstrates the inability of an introduced commercial strain to persist, even within an environment possessing minimal competitive pressure from an indigenous microbiome (perlite growth medium). That being said, I believe certain of the conclusions raised by the authors are not supported by the experimental protocols used in this study.  These are noted in the specific comments below and should be addressed in a revised version of this manuscript.

Specific comments:

Abstract:

Page 1, Line 19.  Where “soft rot disease” is noted, it would be of benefit to include the name of the causal agent being studied (Pectobacterium) at this point in the abstract.

Page 1, line 23.  The authors suggest that a “synergistic approach combining several treatments” will be necessary to control this disease; however, there was no specific examination of this need in the discussion component of the manuscript.  While I am totally in agreement with the premise, if there is no further insight into the proposition stated in the discussion or conclusion it would be appropriate to exclude this from the abstract.

Alternatively, findings concerning the interaction between the pathogen and the ‘beneficial’ strains observed in vitro would be a strong addition to the abstract and perhaps even a bit novel.

Introduction:

Page 1, line 36. I do not recall soil fumigation ever demonstrating control (or being use for control) of Pectobacterium.  Are there any such reports?  If not, perhaps no need for this statement...could you reference its use for any specific soilborne pathogen of this plant group?

Page 2, line 76-77. Perhaps clarify to state that ‘use of the biocontrol products would be associated with altered composition of soil bacterial communities’

Materials and Methods:

Page 3, line 114.  What was the time duration for the soaking of bulbs in the treatment solutions?

Page 3, line 117.  What were the parameters that were measure and was any attempt made to monitor disease control? While certain aspects are mentioned in the results regarding these assessments, it is appropriate to cite these activities in the methods.

Page 4, line 161.  What growth medium was used in the agar plate assays?

Page 5, lines 201-205.  While greater “relative abundance” of the pathogen was ascertained in the perlite treatment, the actual density of the bacterium was not determined.  Given that specific strains of Pectobacterium brasiliense were used in this study it would have been logical to monitor this bacterium through culture-based methods or qPCR. Such methodology has been reported and is available in the literature. As such, the statement needs to be edited and effort should be made to refrain from indicating that a higher level of the pathogen was detected at this point and throughout the manuscript.

Page 7, lines 224-226. Were there any statistical differences across treatments in the number of OTUs observed?

Page 7, figure 4. Due to the limited font size of the text, the axis designations really cannot be discerned; perhaps stack the figures vertically and expand each; or cut and paste a text box to replace the visualization program generated text.

Page 7, line 258. Again, this is an assessment of relative abundance rather than actual density.  Thus, it cannot be stated that a “higher presence of Bacillus genus” was observed as this infers a difference in absolution density.

Discussion:

Page 11, line 374.  I am not certain as to what is meant by the text “This result indicates poor inhibition of bacteria in the root environment…”  Does this infer that the environment is not supportive of bacterial growth?  Please clarify or state in a more definitive manner.

Page 11, line 375.  What is meant by “a lower quality fo sequencing compared to…”? Are you attempting to state that the perlite environment resulted in an inferior quality of extracted DNA that inhibited the subsequent sequencing reaction?  Again, please clarify.

Page 11, line 376-378. The assumption put forth here by the authors is that increasing species richness is correlated with reducing pathogen activity. This is by no means a universal observation and in fact disease suppression driven by a treatment-induced transformation of the soil microbiome has more often been associated with reduced diversity; Rather the response is correlated with the amplification of specific microbial functions.  In addition, the data acquired in this study failed to assess the overall quantity of microbial biomass carried by the two different growth media.  It is likely that the initial ‘bacterial load’ in the perlite medium was substantially lower than the ‘soil’ and thus what was observed was merely a difference in “competitive exclusion” capacities of the two media. I believe a re-examination or re-writing of this section is appropriate.

Page 12, lines 389-391.  In this instance, as noted above, what is likely being observed is a reduced function/operation of the process termed “competitive exclusion”.  That is, you added a pathogen but the overall initial microbial load supported by the perlite was low compared to the 'soil' substrate. Thus, the elevated competitive environment experienced in the “soil” substrate diminished the ability of the pathogen to thrive.  It would have been of great value to determine bacterial load in these two substrates prior to addition of the pathogen.  In addition, levels of the pathogen were not determined in this study…relative abundance was assessed not soil density and thus the text should be edited to state as such.

Page 12, line 424. With regard to the “Two different Bacillus subtilis products…”:  There are important conclusions that were avoided in the discussion. These microbial products failed to provide disease control in the study reported here and in many, if not most, instances failed to enhance plant productivity even in a growth medium that lacked a robust microbiome.  Those employed in this study were highly non-competitive environments.  Were these products tested in a real soil where intact microbiomes actually existed, these products likely would have been undetectable from any perspective whether direct survival or indirectly through assessment of plant impact.  It would be of benefit if such a view were described or at least considered in this discussion.

Page 13, line 443-445.  Precisely!  Thus, direct assessment of pathogen density would greatly strengthen the findings reported herein.

Conclusion:

Page 13, lines 467-469. It may be just as likely, if not more so, that the difference simply in microbial density between these two systems was responsible for the observed outcome and not differences in bacterial diversity.  The statement as currently put forth is not warranted nor appropriate given that no such assessment on microbial densities resident to these two growth media prior to pathogen inoculation was provided.  In addition, other components of the microbiome that influence the overall robustness of competition encountered by the introduced pathogen (including fungi, protozoa, archaea, etc) were not assessed in this study.  These limitations warrant a re-stating/editing of this statement.

Author Response

Reviewer 1.

Comments and Suggestions for Authors

Overall, this manuscript was a pleasure to read.  It is a well-organized and clearly written document.  The value of bio-stimulants and microbial inoculants as a solution for the control of plant diseases or enhanced plant nutrition is a recurring question.  As such, this work directly addresses that question and is one of the few studies willing to report the lack of efficacy…even though the vast majority of trials have demonstrated ‘failure’ but with the caveat that ‘additional study’ will lead to success.  Likewise, the report of a lack of effect of the inoculant on the soil microbiome is of value and again demonstrates the inability of an introduced commercial strain to persist, even within an environment possessing minimal competitive pressure from an indigenous microbiome (perlite growth medium). That being said, I believe certain of the conclusions raised by the authors are not supported by the experimental protocols used in this study.  These are noted in the specific comments below and should be addressed in a revised version of this manuscript.

Dear reviewer 1, your overall view of our manuscript has lifted our spirits in regards to the long research and work that was done to achieve it, and we are grateful for your support and excellent commentary to make this manuscript even better. Your comments have been most enlightening and it was with great satisfaction to execute the revisions and provide a more scientifically coherent manuscript. The most meaningful subject that arose was the impression that we inoculated the plants in the greenhouse, however, in fact, the disease developed naturally and was not introduced by us. This issue made us look for any place that could mislead the reader. In addition, your comments were most supportive and informative which enabled us to make the revision without changing the overall concept of the manuscript. We are thankful for your effort and hope you will find the changes made to be in line with your comments and your professional approval.

Specific comments:

Abstract:

  1. Page 1, Line 19.  Where “soft rot disease” is noted, it would be of benefit to include the name of the causal agent being studied (Pectobacterium) at this point in the abstract.

Agree. The name of the pathogen has been added.

  1. Page 1, line 23.  The authors suggest that a “synergistic approach combining several treatments” will be necessary to control this disease; however, there was no specific examination of this need in the discussion component of the manuscript.  While I am totally in agreement with the premise if there is no further insight into the proposition stated in the discussion or conclusion it would be appropriate to exclude this from the abstract.

As suggested this remark has been excluded and the sentence has been changed as such: We suggest a density-based and functional analysis in the future, to study the specific interactions between plants, soil type, soil microbiota, and relevant pathogens.

  1. Alternatively, findings concerning the interaction between the pathogen and the ‘beneficial’ strains observed in vitro would be a strong addition to the abstract and perhaps even a bit novel.

Agreed. The following sentence has been added to line 22 of the abstract: In addition, the microbes cultured from these products, could not directly inhibit Pectobacterium growth in vitro.

Introduction:

  1. Page 1, line 36. I do not recall soil fumigation ever demonstrating control (or being used for control) of Pectobacterium.  Are there any such reports?  If not, perhaps no need for this statement...could you reference its use for any specific soilborne pathogen of this plant group?

Correct, the word fumigation is misused and is a mistake in translation from Hebrew. The treatment recommended by the Israeli ministry of agriculture advisors (in official reports and regulations) is the use of Metam-sodium either by spray or administered by driplines under a cover (used for solarization). These sources are in Hebrew but added scientific references to the use of metam sodium as a replacement for the harmful methyl bromide products. The word “fumigation has been replaced in the manuscript with soil treatments.

  1. Page 2, line 76-77. Perhaps clarify to state that ‘use of the biocontrol products would be associated with an altered composition of soil bacterial communities

The sentence has been changed as suggested.

Materials and Methods:

  1. Page 3, lines 102, 107. 

We added on line 102: All soft rot disease incidents in the greenhouse or on the bulbs occurred from a natural infestation of the bulbs or the growing media. On line 107: "…or plated for dual culture assays on minimal medium (MM)…"

  1. Page 3, line 123. What was the time duration for the soaking of bulbs in the treatment solutions?

For a period of 15 min. This data was added to the text.

  1. Page 3, line 117.  What were the parameters that were measure and were an attempts made to monitor disease control? While certain aspects are mentioned in the results regarding these assessments, it is appropriate to cite these activities in the methods.

Corrected on lines 130-134. As seen in the results the disease was measured at the end of the season by indexing bulbs' health and the disease index created for that matter. An addition in line 130 describes the specific disease index based on bulbs symptoms. "For health parameters, the severity of natural occurring soft rot disease in bulbs/tubers per treatment was assessed using the following index: (1) completely healthy; (2) slightly soft; (3) water-soaked spot of soft rot; (4) half of the bulb is rotten; (5) bulbs are completely rotten. For growth parameters, the number of new bulbs and the cumulative weight of bulbs were measured".

  1. Page 4, line 161.  What growth medium was used in the agar plate assays?

Minimal media as described in section 2.2 of materials and methods, we clarified and added the information (line 107) that this media was used for the dual culture assays.

  1. Page 5, lines 201-205.  While greater “relative abundance” of the pathogen was ascertained in the perlite treatment, the actual density of the bacterium was not determined.  Given that specific strains of Pectobacterium brasiliense were used in this study it would have been logical to monitor this bacterium through culture-based methods or qPCR. Such methodology has been reported and is available in the literature. As such, the statement needs to be edited and effort should be made to refrain from indicating that a higher level of the pathogen was detected at this point and throughout the manuscript.

Corrections were made to this section avoiding any declaration of actual levels. The terminology has been changed to “rate” or “relative rate”. Changes were made throughout the manuscript.

  1. Page 7, lines 224-226. Were there any statistical differences across treatments in the number of OTUs observed?

Yes, as written, the biggest difference was between the same treatment (water control) and plant type (O. dubium) in the two different media. To clarify this issue, letters representing the significantly different groups were added to figure 3, so the difference will be more visual.

  1. Page 7, figure 4. Due to the limited font size of the text, the axis designations really cannot be discerned; perhaps stack the figures vertically and expand each, or cut and paste a text box to replace the visualization program-generated text.

The figures were stacked vertically and enlarged as suggested.

  1. Page 7, line 258. Again, this is an assessment of relative abundance rather than actual density.  Thus, it cannot be stated that a “higher presence of Bacillus genus” was observed as this infers a difference in absolution density.

The word presence was altered to “relative abundance”.

Discussion:

  1. Page 11, line 374.  I am not certain as to what is meant by the text “This result indicates poor inhibition of bacteria in the root environment…”  Does this infer that the environment is not supportive of bacterial growth?  Please clarify or state in a more definitive manner.

Yes, the text was changed to:  " This result indicates that the root environment in the perlite substrate is not supportive of bacterial growth [43], and as a result, the quality of extracted DNA for the subsequent sequencing reaction was lower, compared to the substrate containing organic matter".

  1. Page 11, line 375.  What is meant by “a lower quality of sequencing compared to…”? Are you attempting to state that the perlite environment resulted in an inferior quality of extracted DNA that inhibited the subsequent sequencing reaction?  Again, please clarify.

The line has been changed as above.

  1. Page 11, lines 376-378. The assumption put forth here by the authors is that increasing species richness is correlated with reducing pathogen activity. This is by no means a universal observation and in fact, disease suppression driven by a treatment-induced transformation of the soil microbiome has more often been associated with reduced diversity; Rather the response is correlated with the amplification of specific microbial functions.  In addition, the data acquired in this study failed to assess the overall quantity of microbial biomass carried by the two different growth media.  It is likely that the initial ‘bacterial load’ in the perlite medium was substantially lower than the ‘soil’ and thus what was observed was merely a difference in “competitive exclusion” capacities of the two media. I believe a re-examination or re-writing of this section is appropriate.

This assumption was in regards to the benefit of organic amendments to the growing media for disease suppression as cited at the end of the sentence referred. In accordance with your comment text has been added and as follows:

"It is likely that the initial ‘bacterial load’ in the perlite medium, was substantially lower than that of the planting mix substrate, and thus responsible for the difference in “competitive exclusion” capacities of the two media. In literature, there has been evidence that increasing the amount of organic material in the substrate may positively affect plant health and reduce pathogenic inter-actions”.

  1. Page 12, lines 389-391.  In this instance, as noted above, what is likely being observed is a reduced function/operation of the process termed “competitive exclusion”.  That is, you added a pathogen but the overall initial microbial load supported by the perlite was low compared to the 'soil' substrate. Thus, the elevated competitive environment experienced in the “soil” substrate diminished the ability of the pathogen to thrive.  It would have been of great value to determine the bacterial load in these two substrates prior to the addition of the pathogen.  In addition, levels of the pathogen were not determined in this study…relative abundance was assessed not soil density and thus the text should be edited to state as such.

It is most important to state that the disease observed in the greenhouse assays occurred naturally and there was no artificial inoculation of plants with a specific pathogen. This supports the essential need for a solution to control soft rot disease during ornamental geophyte cultivation, as seen in both growing seasons. In the current study we only evaluated symptoms of soft rot disease during the growing season and in the bulbs at the end of the season. The in vitro, dual culture experiments were performed with P. brasilense that have been isolated at the lab in a previous study.  

We agree it would have been of great value to determine Pectobacterium load in these two substrates during the growing season, however, this goal is not doable any longer. 

Since bacterial density was not measured, we have changed the manuscript to clarify this in the conclusions: "Apparently, the elevated competitive environment in the planting mix reduced the relative abundance of the pathogen in this medium. Since bacterial load was not determined, we could only refer to the relative abundance of the different OTUs. The occurrence of soft rot symptoms was higher in perlite, as observed in the greenhouse and on the bulbs, suggesting that a microbiome that contains little organic matter may be less suppressive to pathogen development".

  1. Page 12, line 424. With regard to the “Two different Bacillus subtilis products…”:  There are important conclusions that were avoided in the discussion. These microbial products failed to provide disease control in the study reported here and in many, if not most, instances failed to enhance plant productivity even in a growth medium that lacked a robust microbiome.  Those employed in this study were highly non-competitive environments.  Were these products tested in real soil where intact microbiomes actually existed, these products likely would have been undetectable from any perspective whether direct survival or indirectly through assessment of plant impact.  It would be of benefit if such a view were described or at least considered in this discussion.

Correct, there has been much work with these products on Rhizoctonia in potatoes and Fusarium in tomatoes and were deemed successful, as claimed by the manufacturer. Though stated in the introduction, this information has been added to the discussion in lines 438-440: "Both of these products were tested on the fungal pathogens Rhizoctonia in potatoes and Fusarium in tomatoes, both in real soil grow-ops, and were deemed successful, as claimed by the manufacturer".

  1. Page 13, line 443-445.  Precisely!  Thus, direct assessment of pathogen density would greatly strengthen the findings reported herein.

We fully agree, however at the current stage this is no longer possible. Accordingly we have clarified in the last sentence before conclusions:    "… no effects of the beneficial bacteria on the plants were observed (as observed by evaluation of soft rot symptoms), raising the question if they were present at all".

Conclusion:

  1. Page 13, lines 467-469. It may be just as likely, if not more so, that the difference simply in microbial density between these two systems was responsible for the observed outcome and not differences in bacterial diversity.  The statement as currently put forth is not warranted nor appropriate given that no such assessment on microbial densities resident to these two growth media prior to pathogen inoculation was provided.  In addition, other components of the microbiome that influence the overall robustness of competition encountered by the introduced pathogen (including fungi, protozoa, archaea, etc.) were not assessed in this study.  These limitations warrant a re-stating/editing of this statement.

It is true that microbial density was not measured and that other unknown factors as suggested may have been crucial to ascertain the results presented. Accordingly, we have included at the end of the conclusions section the following comments:  

Lines 502-505: "Planting-mix medium shows to be superior for both tested geophyte crops, in terms of both the richness of the bacterial communities and the repression of soft-rot disease as observed in the microbiome analyses and soft rot disease assessment in the greenhouse and in bulbs, respectively.

Line 510: "Assessment of microbial densities in the soil or growth media prior to planting and understanding of other components of the microbiome (including fungi, protozoa, archaea) would improve the understanding of the overall robustness of competition encountered by a pathogen".

Reviewer 2 Report

The manuscript "Root-associated microbiomes, growth, and health of ornamental geophytes treated with commercial plant growth-promoting products" its a very well presented article.

My comments :

  1. IN Materials & Methods: The authors must to present the experiment design better. For example , the replication, the main factors etc. 
  2. At figures2 & 3 I suggest to authors to add letter for comparisons between treatments (water, ecosense....etc)
  3. At Figures 7 & 8 . I suggest to authors to replace the asterisks (*) with Letters (a, b etc for significant differences)

Author Response

Reviewer 2.

Comments and Suggestions for Authors

The manuscript "root-associated microbiomes, growth, and health of ornamental geophytes treated with commercial plant growth-promoting products" is a very well presented article.

Dear reviewer 2, we thank you for your positive review of our manuscript. We appreciate the useful comments, which we have used to improve our manuscript. We have included within our comments and the changes we have made accordingly. 

  1. IN Materials & Methods: The authors must present the experiment design better. For example, the replication, the main factors, etc. 

It is not very clear from the comment, to which experimental setup the reviewer refers. Thus, according to the comment the number of plants in each growing season and the number of replication plates in the dual culture assays were added to section 2.3, and to section 2.5, respectively.

  1. In figures 2 & 3 I suggest to authors add letters for comparisons between treatments (water, ecosense....etc).

These figures are means and percentiles of thousands of sequences generated by the microbiome taxonomic analyses. In figure 2 we only wanted to show any presence and relative abundance, which by the method of collecting this data could not be compared for statistical significance between groups.

We agree that in figure 3 the differences could contribute to the understanding of the data. To clarify this letters representing the significantly different groups were added to figure 3, so the significant difference will be visual.

  1. At Figures 7 & 8. I suggest to authors replace the asterisks (*) with Letters (a, b, etc for significant differences)

In this research, our aim was not to compare the treatments with each other, but rather to see if they display any significant effect relative to the water control. We clarified this issue, in section 2.6 of materials and methods in line 186 as such: " Student’s t-test was performed, to test each treatment relative to the water control".
